# Emerging Roles of Natural Compounds in Osteoporosis: Regulation, Molecular Mechanisms and Bone Regeneration

**DOI:** 10.3390/ph17080984

**Published:** 2024-07-25

**Authors:** Sidra Ilyas, Juni Lee, Donghun Lee

**Affiliations:** Department of Herbal Pharmacology, College of Korean Medicine, Gachon University, 1342 Seongnamdaero, Sujeong-gu, Seongnam-si 13120, Republic of Korea; sidrailyas6@gachon.ac.kr (S.I.); lgl5278@gachon.ac.kr (J.L.)

**Keywords:** natural compounds, osteoporosis, immune cells, ROS, epigenetics

## Abstract

Bone health is a critical aspect of overall well-being, and disorders such as osteoporosis pose significant challenges worldwide. East Asian Herbal Medicine (EAHM), with its rich history and holistic approach, offers promising avenues for enhancing bone regeneration. In this critical review article, we analyze the intricate mechanisms through which EAHM compounds modulate bone health. We explore the interplay between osteogenesis and osteoclastogenesis, dissect signaling pathways crucial for bone remodeling and highlight EAHM anti-inflammatory effects within the bone microenvironment. Additionally, we emphasize the promotion of osteoblast viability and regulation of bone turnover markers by EAHM compounds. Epigenetic modifications emerge as a fascinating frontier where EAHM influences DNA methylation and histone modifications to orchestrate bone regeneration. Furthermore, we highlight EAHM effects on osteocytes, mesenchymal stem cells and immune cells, unraveling the holistic impact in bone tissue. Finally, we discuss future directions, including personalized medicine, combinatorial approaches with modern therapies and the integration of EAHM into evidence-based practice.

## 1. Introduction

Bone, a complex tissue, has a unique structure that requires permanent renewal for its health, balance and integrity. Bone remodeling, a continuous process, occurs at specific sites called basic multicellular units (BMUs) and hinges on two critical processes: osteogenesis and osteoclastogenesis [1,2]. The delicate balance between osteogenesis and osteoclastogenesis ensures optimal bone turnover, a cornerstone of overall skeletal health.

Bone remodeling generally occurs in the following steps: in the initiation phase, the immature precursor cells that have the potential to become bone-resorbing osteoclast are activated by signaling molecules, selected to differentiate into mature active osteoclast and secrete enzymes required for bone resorption [3]. The inversion phase begins when osteoclast activity is halted so that the osteoclast undergoes apoptosis. In the terminal phase, bone-forming osteoblasts are proliferated, differentiated by bone morphogenetic protein (BMP) and synthesizing the bone matrix to promote mineralization by calcium and phosphate ions [4].

Osteoblasts and osteoclasts are mainly derived from multipotent mesenchymal stem cells (MSC) and hematopoietic cells (HSCs) of the monocyte/macrophage lineage, respectively [5,6]. East Asian Herbal Medicine (EAHM) plays a significant role by enhancing osteoblast proliferation/differentiation, thereby amplifying bone formation. On the other hand, osteoclasts are tightly regulated by EAHM bioactive molecules. By inhibiting osteoclast activity, EAHM can help maintain bone density and strength. Research into how EAHM modulates these cellular events holds immense potential for uncovering novel therapeutic strategies to combat bone-related conditions. Signaling pathways are the main conductor in the intricate mechanism of bone remodeling, balancing bone formation/resorption. Among these, the Wnt/β-catenin, BMP/Smad, MAPK and RANKL/OPG pathways are the leading role players [7,8,9]. EAHM influences osteoblasts/osteoclasts proliferation and differentiation by activating/inhibiting the signaling pathways. 

Chronic inflammation interrupts bone remodeling, disturbing the balance between bone formation/resorption. EAHM emerges as a maestro, restoring bone harmony. Under a decreased estrogen level, increased RANKL levels can upregulate the production of NF-κB, pro-inflammatory cytokines (IL-1, IL-6, TNF-α, IL-1β) and ROS to secrete more RANKL produced by osteoblast to downregulate OPG expression. Cytokines and prostaglandins boost osteoclastogenesis by increasing/decreasing RANKL and OPG, respectively. EAHM compounds serve as potent anti-inflammatory modulators within the bone microenvironment. Anti-inflammatory polyphenols and their derivatives downregulate the over-activation of NF-κB, a key transcription factor. Additionally, EAHM inhibits cyclooxygenase2 (Cox2), an enzyme that fuels inflammation and reduces the production of pro-inflammatory cytokines (TNF-α and IL-1β), which act as destructive messengers. EAHM can promote osteoblast development and accelerate the formation of bone by activating the transforming growth factor-β (TGF-β), vascular endothelial growth factor (VEGF), osteoprotegerin (OPG) and bone morphogenetic proteins (BMP). These actions effectively preserve osteoblast viability and function and promote osteoblast proliferation, ensuring a robust group of bone-forming cells. By safeguarding osteoblasts from apoptosis and stimulating matrix synthesis, EAHM compounds contribute directly to bone tissue growth. 

The regulation of bone biomarkers is a critical aspect of maintaining bone health and preventing osteoporosis (OP), as these markers serve as indicators of bone formation and resorption dynamics. Alkaline phosphatase (ALP), osteocalcin (OC), osteoprotegerin (OPG), runt-related transcription factor 2 (Runx2) and osterix (Osx) are the key regulators for enhanced osteoblast activity [10], whereas osteoclast-associated markers include tartrate-resistant acid phosphatase 5b (Trap5b), cathepsin K (CtsK), matrix metallopeptidase 9 (MMP9), collagen alpha 1 (Col1A1) and receptor activator of nuclear factor-κB ligand (RANKL) [11]. 

Genes hold the blueprint for the biological system, but its epigenetics, the layer of chemical instructions on top of the genes, decide how they play a significant role. EAHM emerges as a potential modulator of epigenetic processes. One key way EAHM influences bone health is by interacting with DNA methylation. EAHM compounds can alter methylation patterns, potentially promoting the activity of genes crucial for osteogenic pathways. Another tactic employed by EAHM involves histone modifications, which are the proteins that package DNA. EAHM can influence acetylation/deacetylation, which act as switches to loosen/tighten the DNA packaging. This, in turn, regulates how accessible the genes are to cellular machinery. By influencing these switches, EAHM can fine-tune gene transcriptions, ensuring the right instructions are readily available for bone-building processes. Understanding how EAHM interacts with these epigenetic marks opens exciting possibilities. It paves the way for personalized approaches to bone regeneration and health, tailoring treatment based on an individual’s unique epigenetic landscape. By working with regulatory mechanisms, EAHM offers a promising avenue for promoting bone health and potentially treating bone-treating conditions. 

As far as the bone microenvironment is concerned, a network of specialized cells, such as osteocytes, osteoblast, osteoclast, mesenchymal stem cells (MSCs) and hematopoietic cells (HSCs), as well as immune cells (macrophages, T cells and dendritic cells), are constantly communicating and collaborating. Osteocytes, the most abundant cell type in bone, reside within the matrix and communicate with osteoblasts and osteoclasts, ensuring a balanced and efficient remodeling process. EAHM compounds can influence their function by modulating signaling pathways. By preserving osteocyte viability, EAHM helps maintain their crucial role in mechano-sensing, the ability to detect physical forces on the bone and adapt accordingly, leading to bone homeostasis. MSCs are the multipotent powerhouses within the bone microenvironment that hold immense potential for bone regeneration. MSC differentiates into osteoblast under the influence of hormones, cytokines and various growth factors. By creating a supportive environment for MSCs to thrive, EAHM can promote bone healing and regeneration. EAHM compounds can stimulate the proliferation of these stem cells and nudge them towards differentiating into specialized bone-forming osteogenic lineages. Moreover, immune cells play an active role in bone remodeling. A delicate balance exists between pro/anti-inflammatory signals, both of which affect bone health. By regulating the immune system, EAHM can indirectly influence osteoclast activity and osteoblast function, ultimately promoting a healthy bone microenvironment. The influence of EAHM extends beyond individual cell types; it shapes the intricate dynamics of osteocytes, MSCs and immune cells, ultimately contributing to bone tissue repair and resilience. 

In this review, we explore how EAHM compounds influence bone health, examining their effects on bone cell activity, inflammation, and regeneration. It explores exciting possibilities for personalized medicine and the mainstream integration of EAHM in bone care. 

## 2. Osteogenesis and Osteoclastogenesis Modulation

### 2.1. Promoting Osteoblast Differentiation

EAHM offers a unique approach to promoting bone health by influencing osteoblasts, which serve as a strong foundation for bone formation. MSCs have the potential to be divided into various cell types. EAHM bioactive molecules act as key activators with a specific agenda, nudging these versatile cells toward becoming osteoblasts. They turn on essential transcription factors such as Runx2, core binding factor α1 (Cbfα1) and osterix (Osx), which are the master switches, directing the MSCs to commit to the osteogenic lineage. These transcription factors control osteogenic differentiation by regulating the expression of *ALP*, *OCN* and *OPN* genes that promote the mineralization of bone nodules [12]. It has been investigated that complete *Runx2* gene deletion in mice inhibits osteoblast growth, while a single copy disruption can cause cleidocranial dysplasia [13]. Tanshinone IIA (TSA), a lipophilic compound from *Salvia miltiorrhiza*, also stimulates osteogenic differentiation through Runx2 activation via the ERK1/2 pathway [14,15]. The success of this persuasion is evident by the increased expression of osteogenic markers (ALP and OCN). This indicates a rise in the number of mature osteoblasts, and with more osteoblasts, bone formation receives a significant boost. By promoting osteoblast development/differentiation and preventing osteoclastogenesis, emodin from *Polygonium multiflorum* controls bone remodeling by enhancing the expression of ALP and OCN [16]. Emodin enhances the expression of Runx2 (master regulator of osteogenesis) and its downstream targets, osteocalcin (OCN), to ensure that osteoblasts differentiate properly and synthesize the necessary matrix components. It inhibits osteoclast differentiation by RANKL and suppresses NF-κB activity (Table 1). Additionally, EAHM compounds such as Ugonin K activate Osx, another transcription factor that guides osteoblasts toward a mature functional state. The mechanism involves the interaction of Ugonin K with an estrogen receptor that induces the expression of ALP and an intrinsically disordered protein, OCN, thereby activating the non-classical c-Src signaling pathway. It also upregulates the expression of estrogen receptors and phosphorylate tyrosine protein kinase, c-Src [17]. 

### 2.2. Stimulation of Bone Matrix Synthesis

Bone is a complex composite material. Collagen, proteoglycans and other structural proteins are the building blocks that give bone its shape and support. EAHM compounds act as facilitators, encouraging the production of essential extracellular matrix components with bone tissue. By promoting their synthesis, EAHM contributes to a more robust and well-structured matrix. These newly formed building blocks need to be cemented together for true strength. EAHM influences the process of mineralization, which involves the deposition of hydroxyapatite crystals (calcium phosphate) within the bone matrix. Hydroxyapatite crystals solidify the collagen matrix and enhance its mechanical properties, making it more resistant to stress and fractures. EAHM compounds play a crucial role in enhancing the synthesis of essential extracellular matrix components by osteoblasts. Collagen type I (Col1A1), osteopontin (OPN) and osteocalcin (OCN) are the building blocks of this matrix. EAHM molecules act as blueprints, upregulating the expression of these proteins. Collagen provides the essential scaffolding, while OPN and OCN regulate the deposition of minerals to solidify the structure. OPN modulates the immune system by acting as a pro-inflammatory cytokine by increasing the expression of cytokines and bone matrix degradation enzymes [62]. By promoting robust matrix synthesis, EAHM contributes to a stronger and more resilient bone structure. EAHM compounds influence mineralization-related proteins such as ALP and OPN, which place the groundwork for hydroxyapatite formation. By promoting mineralization, EAHM ensures that the bone framework achieves its optimal hardness and rigidity. One such compound is Avicularin, which suppresses cartilage extracellular matrix degradation and inflammation via TRAF6/MAPK activation [63]. Quercetin (Que), a common flavonoid found in fruits and vegetables, can also upregulate the expression of genes *ALP*, *OPN* and *Col1A1*, as well as transcription factors Runx2 and Osx, to stimulate bone matrix formation and enhance differentiation of MSC to become osteoblast [40]. The mechanism involves the activation of the Wnt/β-catenin signaling pathway by inhibiting the degradation of β-catenin (Figure 1). Berberine, derived from *Coptis chinensis*, has been shown to enhance bone formation by suppressing adipocyte differentiation [47].

### 2.3. Anti-Apoptotic Effects on Osteoblasts

Bone formation is a dynamic process, and osteoblasts constantly face challenges such as oxidative stress and inflammation, and these environments can threaten their survival through a process called apoptosis. EAHM molecules activate specific survival pathways within osteoblasts, such as PI3K/AKT, Bcl2 family proteins, and ERK [65]. *Moringa oleifera* leaf extract can protect and enhance osteoblast development upon damage caused by oxidative stress/aging by activating anti-apoptotic PI3K/AKT/FOX1 pathways [38]. These pathways shield, protect and promote osteoblast cell viability by modulating genes, ensuring proper osteoblast function. By preserving these essential cells, EAHM safeguards the ongoing process of bone remodeling and contributes to long-term bone health. EAHM contains secondary metabolites (flavonoids and terpenoids) that can stimulate osteoblast proliferation and differentiation, making a larger pool of osteoblasts that are thus capable of producing more bone tissue for growth and repair. Crocin can attenuate osteoporosis by preserving/protecting osteoblast in rats through the inhibition of RANKL via regulating JNK and NF-κB signaling, ensuring healthy and efficient bone regeneration and a productive osteoblast population [66,67]. 

### 2.4. Inhibition of Osteoclast Formation 

Osteoclasts are developed from macrophage precursor cells under the influence of multiple cytokines, prostaglandins and oxidative stress [35,68]. Pre-osteoclast RANK receptors are present, which bind to their ligand, RANKL, and their binding stimulates osteoclast formation and recruits TNF receptor-associated factor 6 (TRAF6) [69]. The increased expression of osteoclast transcription factors such as a nuclear factor of activated T cytoplasmic 1 cell (NFATc1/c-Fos), RANKL activates nuclear factor κB (NF-κB), macrophage colony-stimulating factor (MCSF), c-Jun N-terminal kinase (JNK), extracellular regulated kinases (ERK) and p38 can lead to osteoporosis [70]. Osteoclast-specific proteins such as cathepsin K (Cat K), matrix metalloprotein (MMP9) and triiodothyronine receptor auxiliary protein (TRAP) are also expressed as a result of enhanced transcription factors expression (Table 1). 

Bioactive molecules within EAHM suppress the differentiation of pre-osteoclasts into mature osteoclasts by downregulating the expression of RANKL and CatK [19,36]. The effectiveness of this intervention reflects the reduced expression of osteoclast-specific genes and proteins such as TRAP, CatK and MMP9, which are the blueprints for the osteoclast’s destructive machinery. Naringenin can efficiently attenuate the expression of TRAP and RANK, which are the major negative factors for bone resorption. Moreover, it promotes the expression of the parathyroid hormone 1 receptor (PTH1R) and activates osteogenesis via the Wnt/β-catenin signaling pathway [24]. Another compound, Triptolide, from *Tripterygium wilfordii* can increase the expression of osteoprotegerin (OPG), which can block the binding of RANKL to its receptor, thus suppressing osteoclastogenesis by increasing osteoclast apoptosis [22,71]. By suppressing RANKL expression and availability, EAHM compounds effectively stop and discourage osteoclastogenesis. *Drynariae fortunei* extract hinders the formation of mature osteoclasts by modulating the expression of ALP, BMP2, Col1A1 and collagenase 1 (Table 1) [35,68]. 

Naringin increases the differentiation of MSC to osteoblast by activating two signaling pathways: Wnt/β-catenin and BMP2. The mechanism involves the binding of naringin to estrogen receptor α, increasing the interaction from FAS to FASL, which induces apoptosis of the osteoclast [51]. It increases the expression of genes *BMP2* and *Runx2* involved in osteogenesis and decreases the genes associated with adipogenesis. Naringin alters the interaction of RANK binding to RANKL, decreases the production of inflammatory cytokines and induces osteoclast apoptosis [52]. Naringin balances the bone metabolism towards an anabolic state by mimicking the effect of estrogen. Curcumin, widely used for the treatment of osteoporosis, can inhibit osteoclast formation and promote osteoblastic survival by the MAPK and OPG/RANK signaling pathways [72,73]. Swertiamarin protects against bone loss by suppressing TNFa, IL-6, iNOS and MMP9 via MAPK (ERK, JNK, p38) signaling pathways (Table 1) [26]. 

### 2.5. Attenuation of Osteoclast Activity

EAHM’s influence not only stops the formation of new osteoclasts but also affects the activity of existing ones. EAHM disrupts their operations in several ways, such as by hindering the formation of the sealing zone and the actin ring, which are crucial structures that allow osteoclasts to adhere to the bone and create resorption pits. Additionally, EAHM molecules interfere with the activity of the proton pump (ATPase) located in the ruffled border of osteoclasts. This pump is a powerful drill that breaks down bone minerals, and an interaction between ATPase and microfilaments is required for osteoclast bone resorption. Fruits and vegetables are rich in Quercetin (Que), which can cause actin ring disintegration in osteoclast-like multinucleated cells (OCLs) in a reversible manner. Que prevents the early differentiation of osteoclast progenitors into pre-osteoclasts and actin ring development in mature osteoclasts [74]. By triggering the Wnt/β-catenin signaling pathway, Neringin reduces osteoclast activity and promotes bone preservation [75]. An EAHM herb, *Acorus tatarinowii*, contains α-asarone (ASA), which can prevent osteoporosis by suppressing osteoclastogenesis via AKT, p38 and NF-κB, followed by the NFATc1/c-Fos signaling pathway [23]. Icariin, found in *Epimedium brevicornum*, promotes the development of osteoblasts while hindering the activity of bone-resorbing osteoclasts [19]. It can also stimulate the cAMP signaling pathway in the primary cilia of osteoblasts, hence promoting bone production in developing rats [76]. Another important herb, *Viburnum lutescens*, produces Hydroxyurosolic acid (HUA), which can inhibit RANKL and osteoclast differentiation by suppressing c-Fos and NF-κB signaling [21]. By disrupting the above functions, EAHM effectively reduces osteoclasts’ destructive power. As a result, the signs of their activity, such as bone resorption pits and erosions on trabecular surfaces (bone’s internal network), become less evident. 

### 2.6. Regulation of Osteoclast Survival

Osteoclasts are non-immortal, and their lifespan is tightly controlled. EAHM influences this process by modulating specific survival pathways within these cells. NF-κB and JNK pathways control pro/anti-apoptotic signals. Crocin isolated from *Crocus sativus* exhibits both pro/anti-apoptotic properties by influencing the expression of pro-apoptotic (Bim, Bad) and anti-apoptotic (Bcl2, BclxL) proteins, hence protecting osteoclasts from the ROS/Ca^2+^ pathway. This regulation ensures osteoclasts contribute to maintaining a healthy balance between bone formation and resorption [41]. The loss of Bcl2 leads to upregulated caspase3 expression and increased apoptosis in osteoclasts via the mitochondrial apoptotic pathway. The most prevalent communicator between osteoclasts and osteoblasts are the osteocytes that promote well-organized and effective bone remodeling. Upon osteocyte apoptosis, ATP is released through the Panx1 channels, which can increase RANKL expression and promote the fusion of osteoclast precursor cells into mature osteoclasts. RANKL and TNF-α in osteoclast enhance BclxL expression to prevent bisphosphonate-induced apoptosis. It has been reported that luteolin from *Reseda odorata* can attenuate glucocorticoid-induced osteoporosis by regulating the ERK/LRP-5/GSK-3β signaling pathway in vivo and in vitro [61]. Similarly, mangiferin inhibits apoptosis in dexamethasone-induced MC3T3-E1 cell lines by activating BMP2/SMAD1 signaling and altering RANKL and OPG levels [32]. Additional studies revealed that overexpression of Bcl2 disrupted osteoblast development, impairing the cell’s capacity to develop and generate bone. It also resulted in the loss of osteocytes, which are significant adult bone cells [77]. The role of Bcl2 subfamily proteins in osteoporosis is very complex and still in infancy, and more research is needed. Quercitrin nanocoated implants dramatically reduced the expression of proteins linked to osteoclasts, such as Trap, RankL, cathepsin K, ATPase and Mmp9, in vivo [78]. 

## 3. Signaling Pathways in Bone Remodeling

### 3.1. Wnt/β-Catenin Pathway (Bone Formation)

The cytoplasmic protein β-catenin is the major modulator in the canonical Wnt pathway, as it is degraded continuously in the absence of the Wnt pathway (Figure 1). Wnt ligands bind to specific receptors (Frizzled) and co-receptors (LRP5/6) on osteoblasts to inhibit the degradation of β-catenin by acting on its cell-surface receptor, leading to the stabilization and nuclear translocation of β-catenin [79]. Once the β-catenin is inside the nucleus, it collaborates with transcription factors such as TCF/LEF; they direct the osteogenic components for bone formation by organizing the regulating genes (*Runx2* and *OCN*). EAHM modulates Wnt signaling, acting as a co-conductor, promoting osteoblast differentiation and subsequent bone matrix synthesis. By potentially enhancing this pathway, EAHM compounds such as resveratrol (RSV) and Quercetin (QUE) promote osteoblast proliferation/differentiation by activating Wnt/β-catenin pathways and NAD^+^ dependent deacetylase (sirtuin 1; SIRT1) [80,81]. Upon activating SIRT1 and ATP, RSV influences two signaling pathways: RANK and Wnt. SIRT1 downregulates RANKL expression to inhibit osteoclastogenesis, whereas Wnt gets activated by RSV to induce osteogenesis, suppressing osteoclast differentiation by attenuating osteoblast apoptosis [82]. The mechanism involves the inhibition of RANKL (released by osteoblasts and essential for osteoclast activation/development) to its receptor, RANK (Figure 1). The binding of RANKL to its receptors initiates NF-κB activation, which is required for accelerated osteoclast differentiation [83]. SIRT1 activation by RSV can suppress RANKL-induced NF-κB activity, thereby suppressing osteoclast differentiation [84]. Through wnt1/β-catenin and Runx2 signaling, another EAHM ingredient, Oridonin (ORI), enhanced MSC differentiation into mature osteoblasts by increasing ALP and calcium nodules. By inhibiting osteoclast production by targeting the proteins associated with osteoclasts, such as TRAP and NFATc1/c-Fos, it increases OPG while decreasing RANKL expression (Table 1). Similarly, Catalpol from *Rehmanniae glutinosa* enhances the osteogenic differentiation of MSC, partly via the activation of the Wnt/β-catenin pathway [18].

### 3.2. Bone Morphogenetic Proteins (BMP) Signaling Pathway (Bone Growth Factors)

BMPs, which belong to the TGF-β superfamily, are the key players in the bone remodeling process. When a BMP ligand binds to its receptor, it activates downstream Smad proteins (Smad1/5/8) by phosphorylating on serine residues and forming complexes with Smad4 [85]. Together, they translocate into the nucleus to regulate the expression of osteogenic transcriptional factors such as Runx2 and Oxs. Runx2 is activated by Smad 1/5/8, thus promoting osteoblast differentiation and bone matrix synthesis. EAHM interacts with TGF-β and BMPs. As BMPs are particularly potent inducers of osteoblast differentiation, EAHM compounds may enhance BMP signaling by increasing the expression of BMP receptors (BMPR1A, BMPR2) on osteoblasts and upregulating the genes involved in BMP signaling. This cross-talk between EAHM and signaling pathways amplifies the overall osteogenic response, leading to more efficient bone formation. Salidroside (SAL), a putative BMP2 activator isolated from *Rhodiola rosea* can increase the bone mineral density by the phosphorylation of Smad1/5/8 and ERK1/2, thus activating the genes that control BMP signaling via *BMP2*, *BMP6* and *BMP7* [33]. EAHM-mediated enhancement of this pathway ensures optimal bone growth by osteoblast differentiation. An indirubin derivative, E738, a potent protein kinase inhibitor, acts as a molecular switch to turn on/off the BMP signaling pathway. Smad proteins are activated upon binding to BMP receptors, and further binding to linker proteins (MH1/2) is required to enter into the nucleus for regulating gene expression. E738 helps Smad proteins facilitate their binding to linker proteins to amplify BMP signaling pathways. On the other hand, E738 targets Smad proteins for ubiquitination and turns off BMP signaling [55].

### 3.3. MAPK Pathways (Diverse Ensemble for Balanced Remodeling)

The mitogen-activated protein kinases (MAPK) are the secondary messengers encompassing diverse ensembles within the bone remodeling. EAHM interacts with MAPK (ERK, JNK and p38) pathways, each playing a distinct role (Figure 1). MAP kinases such as ERK can promote Runx2 phosphorylation once activated and dedicated to osteoblast proliferation and survival, ensuring a healthy pool of bone-building cells. JNK plays a more complex role, influencing both osteoblast differentiation and apoptosis, ensuring a balanced number of cells. P38 MAPK function is versatile, regulating both osteoclastogenesis and osteoblast function, ensuring a balanced performance between generation and resorption. A compound, 23-Hydroxyursolic acid (HUA) from *Viburnum lutescens*, inhibits osteoclast differentiation by suppressing RANKL, MAPKs, NF-κB and NFATc1/c-Fos signaling pathways and activating osteogenic genes [21]. Similarly, Eucalyptol (EU) increases the expression of osteoblast proliferation and differentiation via ERK phosphorylation [49]. 

### 3.4. RANKL/OPG System (Osteoclast Control)

The receptor activator of the nuclear factor κB ligand (RANKL), as well as the osteoprotegerin (OPG) system, regulates osteoclastogenesis. RANKL is responsible for osteoclast differentiation from precursor cells (monocytes/macrophages). OPG binds to RANKL, preventing its interaction with RANK receptors on osteoclasts. By influencing the balance between RANKL and OPG, EAHM helps maintain a harmonious level of bone resorption. EAHM can also modulate the expression of RANKL by osteoblasts and stromal cells. Various EAHM herbs, such as *Tripterygium wilfordii* (Triptolide), can increase the expression of OPG by binding to RANKL, making it unavailable for its receptor, RANK (Table 1) [22]. 

### 3.5. cAMP Pathway (Bone Formation)

This pathway is an emerging player in bone remodeling, as cAMP activates protein kinase A (PKA), a key regulator of osteoblast function. PKA then phosphorylates transcription factor CREB, promoting the expression of osteogenic genes. EAHM compounds such as Icariin from *Epimedium brevicornu* may influence this pathway by affecting the activity of adenylate cyclase, an enzyme that increases intracellular cAMP levels and signaling by promoting BMP2 [45,46]. EAHM’s potential modulation of this pathway offers another avenue for promoting healthy bone growth. *Epimedium brevicornu* extract also showed enhanced bone formation by increasing the expression of *Runx2*, *Osx* and *BMP2* via the activation of the AC10/cAMP/PKA/CREB pathway in rats without affecting bone resorption [34].

## 4. Anti-inflammatory Effects on Bone Microenvironment

### 4.1. Suppression of NF-κB Activation

NF-κB modulates multiple aspects of inflammation and serves as a central mediator of inflammatory responses. When triggered, it activates a cascade of inflammatory pathways. EAHM compounds in the form of polyphenols, flavonoids, phenolic acid, lignans, stilbenes and alkaloids can inhibit NF-κB activation, preventing its translocation to the nucleus. EAHM molecules can reduce the expression of pro-inflammatory cytokines. With the inflammatory signal dampened, inflammation-induced bone loss is significantly reduced. Bone degradation during bone remodeling releases some products that act as danger signals to activate multiprotein inflammasome, NOD-like receptor family pyrin domain-containing 3 (NLRP3), a key player in initiating inflammation to promote osteoclast differentiation. EAHM helps reduce inflammation and acts as a multifaceted tool to regulate NLRP3 inflammasomes. In mouse macrophages, Dioscin induces the expression of ALP, Runx2 and OCN to promote osteogenesis by suppressing NF-κB nuclear transport, NLRP3 inflammasome and reactive oxygen species (ROS) levels [27]. In the same way, *Drynaria fortunei* extract can suppress inflammatory cytokines (IL-1β, IL-18, TNF-α, IL-6, IL-8, NF-κB) as well as NLRP3 inflammasome, Notch1 and caspase1 by modulating SIRT1 in patients with postmenopausal osteoporosis (Table 1) [28]. Tanshinone IIA (TSA) can prevent inflammatory cytokines (IL-6, IL-1β and TNF-α), nitric oxide synthase (iNOS) and subsequent NO production by affecting signaling pathways as revealed in Figure 2 [14,15].

### 4.2. Downregulation of Cox2 and Prostaglandins

Another key player in the inflammatory response is Cox2, an enzyme that churns out pro-inflammatory prostaglandins, particularly PGE2, which acts as an accelerant for osteoclastogenesis, further promoting bone loss. Swertiamarin inhibits Cox2 expression, effectively reducing PGE2 levels. This deciphers to a more subdued inflammatory response and protection against bone loss fueled by inflammation (TNFa, IL-6, iNOS, MMPs). Swertiamarin can also modulate the expression of an apoptotic marker (caspase 3) and RANKL at both the mRNA and protein levels [26].

### 4.3. Modulation of Macrophage Polarization

In a healthy bone environment, a balanced population of pro-inflammatory M1 macrophages and anti-inflammatory M2 macrophages exists. However, chronic inflammation can tip the scales toward M1 dominance. EAHM compounds act as moderators, shifting macrophage polarization towards the beneficial M2 phenotype. These M2 macrophages are responsible for releasing anti-inflammatory cytokines, such as IL-10, promoting tissue repair essential for bone healing and regeneration. Isobavachalcone (ISO) from *Psoralea corylifolia* can inhibit bone loss in osteoporosis upon binding with ERα receptors. The mechanism involves the suppression of MMP9, Cat K, NFATc1/c-Fos and macrophage M1 polarization via blocking of the ERK and NF-κB pathway [53]. Similarly, Avicularin (AL) from *Polygonum aviculare* leaf can inhibit bone loss around implants in an osteoporosis mouse model. The mechanism involves the reduction in LPS-induced inflammatory cytokines and suppression of macrophage M1 polarization via blocking of the NF-κB pathway, as shown in Figure 2 [54].

### 4.4. Reduction of Reactive Oxygen Species (ROS) 

Oxidative stress (OS) can be triggered both by a reactive antioxidative defense system and inflammation, which often generates a storm of free radicals called reactive oxygen species (ROS) that play a regulatory role in osteoclast differentiation. ROS produced as a result of OS is associated with damaging bone membrane tissues, hindering bone mass formation and leading to dysfunctions of bone-associated cells [86]. EAHM’s anti-inflammatory capabilities in the form of diverse bioactive flavonoids and phenolic acids help in reducing oxidative stress. Tetrahydroxystilbene glucoside (TGS) from *Polygonium multiflora* has a protective effect on osteoblastic MC3T3-E1 cells against oxidative stress (OS) and H_2_O_2_ by increasing ALP, OCN and Col1A1 activities [44]. Moreover, it decreases ROS, RANKL and IL-6 induced by H_2_O_2_. Que and RSV also act as free radical scavengers, mopping up these ROS and protecting osteoblasts and osteocytes from oxidative stress [40,87]. By maintaining a healthy redox balance (free radicals and anti-oxidants), Icariin prevents bone loss fueled by oxidative stress and could effectively reduce osteoclastogenesis via the activation of Nrf2 signaling. Icariin inhibited the ubiquitination degradation of Nrf2 by targeting the Cullin3/Nrf2/OH pathway, as shown in Figure 2 [88]. Similarly, to stop estrogen-deficiency-induced bone loss, Corylifol A reduced intracellular ROS levels by inhibiting osteoclastogenesis, attenuating the MAPK/ERK pathway and NFATc1 activation. Corylifol A did not significantly inhibit/promote osteogenic differentiation and mineralization but can be used as a potential new replacement medication for treating osteoporosis caused by low estrogen levels [39]. Curcumin, a popular compound from *Curcuma longa* (turmeric), exhibits anti-inflammatory properties and may promote bone healing [48]. 

### 4.5. Interplay with Immune Cells

The immune system plays a complex role in bone health. EAHM compounds interact with T cells and dendritic immune cells within the bone microenvironment. They act as modulators, influencing the production of cytokines and shaping the overall immune response. By regulating the immune system, EAHM indirectly affects bone health by influencing osteoclast activity and osteoblast function [89,90]. Research showed that Baohuoside I, an active component of the herb *Epimedium brevicornu*, encourages bone marrow MSC development into osteoblasts rather than adipocytes. Due to its strong anti-oxidant/osteoporotic properties, it can modulate immunological function by increasing ALP activity and decreasing cytokine (IL-1b, IL-6, IL-8 and TNF-α) secretion [29]. Similarly, Que alters the expression of ALP and OPN, which can modulate the immune cells by acting on pro-inflammatory cytokines and stimulating the differentiation of MSC to develop into osteoblast [40].

### 4.6. Herbal Formulas with Anti-inflammatory Properties 

The wisdom of EAHM extends beyond a single compound. Specific herbal formulas, such as *Eucommiae cortex*, *Salvia miltiorrhiza* and *Curcuma longa*, have been traditionally used for bone health and fracture healing due to their potent anti-inflammatory properties [91]. These formulas contain a complex blend of bioactive compounds that work synergistically to target various inflammatory pathways within the bone microenvironment. Standardized patent medication approved by the FDA (China) include Xianling Gubao (XLGB), Gushukang (GSK) and Gusongbao, Jintiange (JTG), which are commercially available in the market for the treatment of osteoporosis [92,93,94,95]. Their formula, compositions and doses are fixed. Some herbal compositions have been shown in Table 2. 

### 4.7. Regulation of Bone Turnover Markers 

ALP, a homodimeric metalloenzyme produced by osteoblasts, represents an osteoblast phenotype and differentiation. Elevated ALP levels, potentially influenced by EAHM compounds, suggest increased bone formation. ALP is important because it generates/regulates the homeostasis of inorganic phosphate (Pi), a key building block for hydroxyapatite crystals. This is deciphered to enhanced mineralization, the process by which calcium deposits harden the bone matrix, with hydroxyapatite being a key mineral [96]. Monitoring ALP levels provides valuable insights into osteoblast function and the overall health of bone formation. It has been investigated that osteoblasts isolated from ALP knockout mice can differentiate normally but are unable to make mineralized nodules [97]. Another crucial player is osteocalcin (OCN), a non-collagenous protein synthesized by osteoblasts that serves as a valuable marker for bone activity. Acting as a cornerstone for mineralization, OCN directly influences the strength and integrity of the bone matrix by binding and regulating Ca^2+^ homeostasis. EAHM compounds have the ability to regulate OCN expression, highlighting its potential to influence bone matrix synthesis. Similar to ALP, elevated serum OCN levels tell healthcare professionals to gain a valuable window into the rate of bone formation. On the other hand, osteopontin (OPN) and collagen type I cross-linked C-telopeptide (CTX) linked with non-collagenous proteins paint a picture of bone resorption. CTX arises from the degradation of collagen type I, the main protein component of bone released during the process of bone resorption by osteoclasts. Crystals of hydroxyapatite sit on collagen to reinforce the structure when compressed.

In osteoporotic bone, less cross-linking between collagen fibrils and non-collagenous proteins has been observed, which lowers tensile strength [98]. Moreover, osteoporosis is associated with larger hydroxyapatite crystals, which weaken and increase the risk of fractures [99]. Elevated CTX levels indicate increased bone resorption activity. Interestingly, EAHM compounds may hold promise in reducing CTX levels, potentially by inhibiting osteoclast activity to preserve bone mass. Monitoring CTX helps assess the overall dynamics of bone turnover, ensuring a balanced interplay between formation and resorption. While CTX reflects the breakdown of collagen, the PINP indicates a different aspect. This marker shows the rate of collagen synthesis during bone formation. EAHM compounds may enhance PINP levels, potentially reflecting increased osteoblast function and active collagen production. Elevated PINP suggests that EAHM might be promoting the creation of a new bone matrix, a crucial step towards building stronger bones. By deciphering the language of these markers, EAHM practitioners gain valuable insights into bone health, paving the way for a more targeted approach to bone health management and the effectiveness of EAHM interventions. Increased ALP and PINP, alongside decreased CTX, paint a picture of potentially enhanced bone formation and reduced resorption, leading to a more balanced bone remodeling process and potentially improved bone health.

## 5. Epigenetic Modifications and EAHM-Induced Bone Regeneration

### 5.1. Epigenetic Memory and Cell Fate

Each cell carries detailed information for biological systems written in the language of epigenetics. These memories influence how cells behave and ultimately determine their fate during differentiation (specialization) and trans-differentiation or lineage reprogramming (direct conversion into another cell type). Allyl sulfide increases osteoblast differentiation and bone density by inhibiting the DNA (mitochondrial-mediated Kdm6b/H3K27me3) epigenetic pathway [100]. EAHM’s success in promoting bone regeneration hinges on its ability to control these epigenetics [101].

### 5.2. Trans-Differentiation via Epigenetic Modification

Trans-differentiation is a remarkable feat in which one cell type directly transforms into another by passing the intermediate pluripotent stem cell stage. EAHM compounds can induce this transformation by acting as epigenetic editors, modifying chemical tags on DNA (methylation) and proteins wrapped around DNA (histone acetylation). These edits essentially rewrite the cellular instructions, altering gene expression patterns and leading to a dramatic shift in cell identity. For instance, EAHM might nudge a nearby cell type towards becoming osteoblast. Numerous signaling pathways, including sirtuins, kinases, steroid receptors and cyclooxygenases, are known to be impacted. Such an epigenetic modifier, RSV, alters mesenchymal stem cell expression by activating SIRT1, which promotes osteoblast development by stimulating Runx2 and Osx and inhibits the production of the expression of pro-inflammatory cytokines [102,103]. Hydroxysafflor yellow A (HSYA) from *Carthamus tinctorius* could also epigenetically regulate β-catenin by histone demethylase and promote osteogenesis and bone development in OVX rats [57]. Another anabolic, the antiresorptive epigenetic modulator Sulforaphane, a naturally occurring isothiocyanate, is involved in osteoblast differentiation by increasing DNA demethylation [56]. Rhein isolated from *Rheum palmatum* leaf and its derivative, thioamide (RT), can improve estrogen-dependent bone loss by blocking RANKL-induced osteoclastogenesis by attenuating MAPK pathways, which has shown great therapeutic promise in the treatment of OP by specifically targeting bones resorption [104]. Furthermore, it counteracts OP-induced alterations in femurs, presumably through reducing Dnmt1/Dnmt3a activity and the hypermethylation of the Klotho promoter [58].

### 5.3. Epigenomic Regulation for Bone Regeneration

EAHM compounds can target specific genes, particularly those encoding osteogenic transcription factors. By influencing DNA methylation and histone modifications around these genes, EAHM promotes their expression, essentially turning up bone-building instructions. The process of accelerating osteoblastic development may be regulated by an epigenetic system. Sulforaphane from cruciferous vegetables can promote osteoblast differentiation by promoting the ten-eleven translocation (TET1/TET2)-dependent hydroxymethylation of DNA, which reactivates gene expression [56]. These epigenetic changes ultimately lead to enhanced bone matrix synthesis and mineralization, leading to stronger bone tissue. Histone acetyltransferase p300 is essential for changing the histone acetylation-induced chromatin architecture that affects gene transcription. Research has demonstrated that the inhibition of osteoclastogenesis in bone-derived cells is caused by RSV-mediated SIRT1 interactions with p300, which regulate RANKL activation of the NF-κB axis. Furthermore, SIRT1 deacetylase induced by RSV has been demonstrated to trigger the development of the SIRT1-p300 complex, which deactivates p300 acetyltransferase and lowers NF-κB-p65 acetylation, eventually inhibiting the production and activity of osteoclasts [60]. *Ribes nigrum* extract contains anthocyanins that can enhance osteoblast proliferation by preventing starvation-induced apoptosis by up/downregulating the expression of Bcl2/Bax, respectively. In apoptotic osteoblasts, these anthocyanins also altered the expression of HDAC1/3 and upregulated the expression of SIRT1/3 and PGC-1α [59].

### 5.4. Clinical Implications and Challenges

EAHM-induced trans-differentiation holds immense promise for the future of bone tissue engineering. By manipulating epigenetic marks, researchers can potentially guide the fate of readily available cells toward becoming osteoblasts. This approach offers a safer and more readily available source of cells for bone regeneration therapies. However, significant challenges remain. We need an accurate understanding of the precise epigenetic modifications required for optimal bone regeneration. This necessitates collaboration between EAHM experts, epigeneticists and tissue engineers, combining their unique knowledge to unlock the full potential of EAHM for bone health. Additionally, personalized approaches based on individual epigenetic profiles may further optimize treatment outcomes. 

## 6. Pharmacokinetics and EAHM Compound Delivery to Bone Tissue

### 6.1. Pharmacokinetics 

EAHM compounds have remarkable potential for bone health, but getting these compounds to their target, the bone tissue, is not always straightforward. Several challenges and strategies are involved in delivering EAHM compounds to their target site. Pharmacokinetics is the study of how drugs travel within the biological system. It tracks a compound through four key stages: absorption (entry into the bloodstream), distribution (movement throughout the body), metabolism (modification by the biological system) and excretion (removal from the biological system). This ADME process is crucial for EAHM compounds to reach their target site (the bone tissue). 

### 6.2. Challenges in Bone Tissue Delivery 

Delivering drugs to bones presents unique challenges. (1) Unlike well-vascularized organs, bone has a restricted blood supply, hindering the distribution of EAHM compounds. (2) The dense bone matrix, composed of minerals, acts as a barrier, making it difficult for compounds to penetrate. (3) Osteoblasts, osteoclasts and osteocytes, the cellular residents of bone, form additional hurdles for drug delivery. EAHM utilizes a wide range of compounds. Each has unique physicochemical properties that influence their pharmacokinetics. Understanding these properties is essential for optimizing delivery to bone tissue. 

### 6.3. Absorption and Bioavailability

Oral administration is a common route for EAHM compounds. However, their bioavailability (the amount that reaches systemic circulation) depends upon several factors. The water-soluble compounds are readily absorbed by the gut. Lipophilic compounds can easily pass through cell membranes. The liver can break down compounds before they enter the bloodstream, reducing bioavailability. The friendly bacteria in the gut can also modify the EAHM compounds, affecting their absorption. Once absorbed, EAHM compounds need to find their way to bone. Some key factors that influence their distribution: (1) Compounds tightly bound to proteins in the blood have limited ability to penetrate tissues such as bone. (2) Lipophilic compounds tend to accumulate in fatty tissues, but some can also deposit in bones. (3) Active bone remodeling creates spaces within the bone matrix, facilitating the access of EAHM compounds. Several mechanisms are involved in EAHM compounds that actually get inside the bone tissue. Passive diffusion -occurs when small lipophilic compounds can passively diffuse through the gaps in the bone matrix. Osteoblasts and osteoclast cells express specific transporters (organic anion transporters) that actively guide EAHM compounds into the bones (Active transport). Moreover, some EAHM compounds, such as flavonoids, can bind directly to the hydroxyapatite crystals in the bone matrix, providing a local reservoir. 

### 6.4. Local vs. Systemic Administration

The route of administration significantly affects delivery. Local application of EAHM compounds directly to bone defects offers the advantage of improved bioavailability. However, this targeted approach might not be suitable for all situations. Systemic administration, typically involves oral or intravenous routes, and requires higher doses to achieve therapeutic concentrations to reach distant bone sites throughout the body as compared to local delivery methods. 

### 6.5. Combining EAHM with Biomaterials

Researchers are exploring innovative strategies to enhance the delivery of EAHM compounds. One approach involves incorporating these compounds into biomaterials in the form of scaffolds and nanoparticles. These carriers offer several advantages. (1) Biomaterials can be designed to release the EAHM compounds slowly over time, improving stability and efficacy (Controlled release). (2) Biomaterials can be targeted to specific areas of bone defects, maximizing the local concentration of the EAHM compounds. Studies have shown that EAHM-loaded biomaterials can significantly improve bone regeneration in laboratory and animal models [105]. Hydroxy-safflower yellow A (HYSA)-doped scaffolds have the ability to upregulate the expression of osteogenic genes and promote the differentiation of rabbit BMSCs and bone healing in rats [105]. Treating bone injuries with a combination of technical materials and Chinese herbs works well. Porous composite scaffolds with the osteogenic phytomolecule, icariin, are integrated to support skeletal regeneration in rabbits with difficult osteonecrotic bone [106]. The scaffolds can improve cell adhesion and proliferation, according to a study that examined the viability of using oleuropein (mostly from *Canarium album*) as a cross-linking agent to create 3D porous composite scaffolds for bone tissue creation [107]. 

## 7. Clinical Translation and Challenges

### 7.1. Proving Efficacy and Safety in Humans 

The potential of EAHM, from research labs to clinical trials, presents a unique set of challenges. Clinical relevance is paramount. Researchers need to demonstrate that EAHM interventions are not only effective in improving bone health but also safe for use in humans. This involves conducting rigorous clinical trials with clearly defined endpoints, such as BMD improvement, pain reduction and improved physical function, as well as fracture healing and reduction in non-unions (fractures that fail to heal properly). These are the crucial measures for EAHM effectiveness. Currently, there is no clinical efficacy and safety evaluation system. However, evidence-based methods can be used for the evaluation of EAHM, which still has many concerns [108]. Internationally recognized standards for EAHM should be used to measure efficacy in addition to reporting adverse effects as a safety measure in research studies. Moreover, high-quality trials/research are needed to prove effectiveness, efficacy and safety for quality and content analysis [64]. A recent study highlighted the effectiveness and safety of EAHM in osteoporosis. Three randomized control groups of participants with osteoporosis were analyzed: those who received standard medication, standard medication in addition to Duhuo Jisheng Decoction (DHJSD), or just DHJSD. They measured the degrees of discomfort, pain, side effects and BMD. The findings revealed that DHJSD, along with standard medication, has fewer adverse effects, is more effective and is safer than other controls. Hence, DHJSD has been shown to be more effective than standard medication taken alone [109]. 

### 7.2. Personalizing EAHM

Patients are not a homogenous group. Factors such as genetics, lifestyle habits and underlying health conditions can influence how individuals respond to EAHM. To optimize treatment outcomes, researchers are exploring personalized medicine approaches that tailor EAHM interventions to each patient’s unique needs. 

### 7.3. Standardization of EAHM Formulation

EAHM formulations are often a blend of various herbs and compounds. A significant challenge lies in standardizing these formulations to ensure consistent quality and reproducible results across different batches and practitioners. This requires addressing variations in (1) the quality and potency of EAHM compounds, which can be easily influenced by the source of plants used. (2) Different extraction techniques can yield varying concentrations of active ingredients. (3) Even with the same source material, slight variations can occur between different batches of EAHM formulation. In order for the EAHM formula to be evaluated scientifically, the precise ingredients and quantities used must be disclosed. As EAHM practitioners use different formulae, it is more challenging to pinpoint the precise effects and standardization of particular medicines used [64]. Researchers should ensure a consistent formula, processing/extraction methods, and ratio of ingredients for more reliable reproducible results in order to compare with other research studies [110]. 

### 7.4. Assessing Potential Risks and Side Effects

EAHM compounds are not without potential risks. They may interact with other medications or have off-target effects on the biological system. Rigorous safety assessments are essential to ensure patient well-being. Researchers must be vigilant in monitoring short-term as well as long-term toxicity. EAHM may contain heavy metals (As, Cd, Hg and Pb) that can interfere with diets/drugs as they are loaded with active compounds that can work on multiple targets [111]. The most common adverse side effects of EAHM compounds include gastrointestinal problems, an elevated risk of bleeding, altered hormone levels, allergic responses, drug interactions and acute liver damage. Although the side effects of EAHM compounds are fewer than those of Western medicine, more than half of the research studies do not address and report the potential side effects [112]. Moreover, numerous clinical studies lack follow-ups and do not follow a 3-year standard procedure.

### 7.5. Gaining Regulatory Approval

Regulatory agencies, such as the FDA (Food and Drug Administration) and EMA (European Medicines Agency), play a crucial role in ensuring the safety and efficacy of medical interventions. EAHM, with its traditional and complex nature, often faces additional scrutiny during the approval process. Researchers need to present robust evidence to demonstrate both the safety and effectiveness of EAHM for regeneration. The primary barriers to EAHM research include research methodology and implementation due to lack of knowledge, funding, clinical trials, patient data, disagreements within the research community and non-reproducible results [113]. 

### 7.6. Placebo Effects and Bias 

The power of the mind can be a significant factor in healing. The close patient–practitioner interaction can influence treatment outcomes through the placebo effect. To account for this, researchers employ blinding techniques and study designs to ensure the observed benefits are truly due to the EAHM interventions. Thoroughly planned clinical trials with appropriate randomization protocols and blinding are essential to determine the actual efficacy of EAHM for osteoporosis. Research studies with a broader range of participants are more likely to yield findings that can be applied broadly. 

### 7.7. Ethical Concerns 

Cultural beliefs and informed consent are cornerstones of ethical EAHM practice. Combining EAHM with conventional medicine can sometimes raise ethical dilemmas. Researchers and practitioners need to navigate these complexities with sensitivity and transparency. 

### 7.8. Making EAHM a Viable Option

EAHM can potentially be a cost-effective approach to bone health. However, accessibility varies greatly depending on location and healthcare systems. Integrating EAHM into mainstream healthcare requires infrastructure development and educational initiatives for healthcare professionals. 

### 7.9. Monitoring Long-Term Effects

Bone regeneration is a gradual process. Long-term follow-up studies are necessary to assess the durability of treatment effects and identify potential late-onset side effects associated with EAHM interventions. Although challenges exist, researchers are actively working to bridge the gap between the promise of EAHM for bone health and its practical application in clinical settings. By addressing these hurdles and fostering collaboration between EAHM practitioners, scientists and regulatory bodies, the future of EAHM in promoting bone health appears bright.

## 8. Future Perspective and Concluding Remarks

The future of bone regeneration is associated with EAHM as well as modern medicines. Here are some exciting movements in this collaborative piece:

### 8.1. Personalized Pathways

Imagine an EAHM treatment tailored just for you, considering your unique genetic makeup, lifestyle habits and overall health status. This is the promise of personalized medicine. By incorporating these patient-specific factors, precision medicine ensures an optimal outcome, allowing EAHM to be truly helpful for each individual. 

### 8.2. Combinatorial Approaches 

The treatment strengthens when EAHM joins with conventional therapies in the form of surgery, medication (pharmacotherapy) and biological treatments (biologics). 

### 8.3. Advanced Biomaterials and Tissue Engineering 

Imagine 3D-printed scaffolds infused with EAHM compounds or nanomaterials designed for the controlled release of these powerful remedies. These are just some of the possibilities offered by advanced biomaterials and tissue engineering. By incorporating cutting-edge technology, we can optimize the delivery and efficacy of EAHM compounds for efficient bone regeneration.

### 8.4. Gene Therapy and Epigenetic Modulation 

The future holds exciting possibilities for gene editing techniques such as CRISPR/Cas9, potentially enhancing bone healing. Additionally, EAHM can play a role by influencing epigenetic marks, the chemical instructions that shape gene expression patterns. This opens doors for even more targeted and personalized approaches. 

### 8.5. Clinical Trials and Evidence-Based Practice 

EAHM-based interventions require robust clinical trials. This ensures their effectiveness and safety, building a strong foundation of evidence-based practice. With clear guidelines based on research, healthcare professionals can make informed treatment decisions for their patients.

### 8.6. Education and Integration 

Healthcare professionals need to understand the principles of EAHM. Integrating EAHM education into medical curricula and clinical practice allows for a more holistic approach to patient care. By creating bridges between these two disciplines, we can unlock the full potential of this collaborative approach to bone health. 

In conclusion, by synergizing the time-tested wisdom of EAHM with the cutting-edge insights of modern science, we can revolutionize bone health and patient care. This future symphony holds immense promise for a world where strong and healthy bones are a reality for everyone. 

## Figures and Tables

**Figure 1 pharmaceuticals-17-00984-f001:**
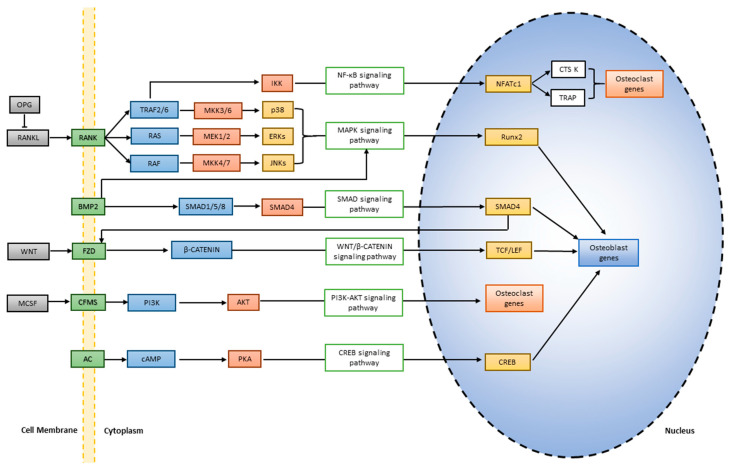
Signaling pathways in osteogenesis and osteoclastogenesis. Osteoblasts produce RANKL that is either neutralized by circulating OPG or triggers a signaling cascade involving TRAF6 and MAPK which activates transcription factors (NFATc1 and NF-κB) for bone resorption. SMAD-dependent BMP signaling involves a complex containing Smad4 which translocate into the nucleus to trigger osteogenesis. SMAD4 also binds to FZD promoter and activates Wnt signaling pathways [64]. Binding of Wnt ligand to frizzled (FZD) receptors initiates Wnt β-catenin signaling pathway. Ligand binding activates adenylate cyclase (AC) that convert ATP to cAMP which ultimately binds to protein kinase A (PKA) to phosphorylate cAMP response element binding protein (CREB) at Ser133 site to control gene expression.

**Figure 2 pharmaceuticals-17-00984-f002:**
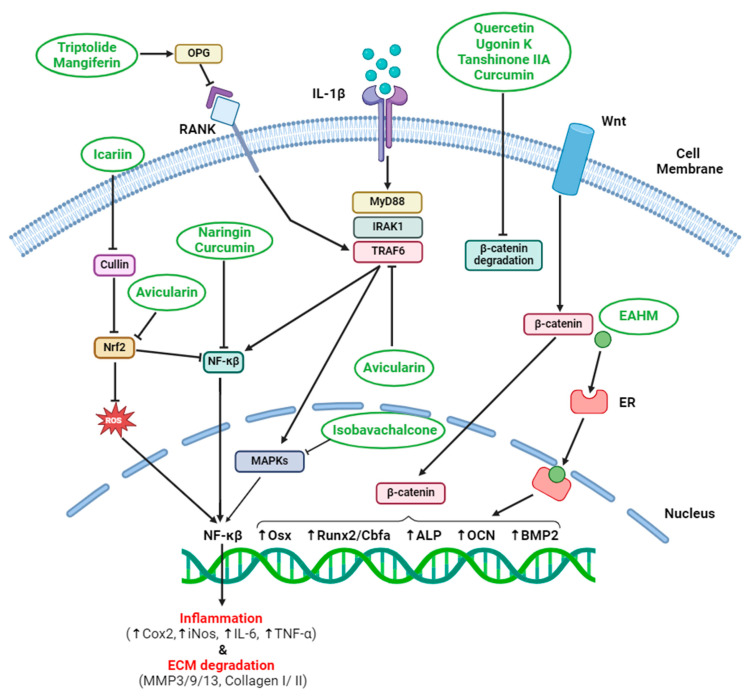
East Asian Herbal Medicine (EAHM) effect on bone to prevent osteoporosis. Osteoblast proliferation/differentiation occur through the activation of signaling pathways such as Wnt/β-catenin, MAPK or by inhibiting the NF-kB and RANKL/RANK. EAHM compounds (green circles) such as Icariin, Triptolide, Mangiferin, Naringin, Curcumin, Quercetin, Ugonin K, Tanshinone IIA, Avicularin and Emodin can alter several targets to exhibit anti-osteoporosis effects. Inverted T marks (⊥) denote inhibition whereas arrows (↓) indicate the activation (Figure generated by Biorender.com).

**Table 1 pharmaceuticals-17-00984-t001:** Role of Natural Chinese medicine and their anti-osteoporotic effects.

Target	EAHM	Ingredient	Mechanism	Pathways	Study	Outcome	Ref.
RANKL	*Rehmanniae glutinosa*	Catalpol	Increases ALP, Ca and MSC differentiation to osteoblast	Activated Wnt/β-catenin signaling	Pre-clinical	Promote osteogenesis	[18]
	*Rheum palmatum*	Emodin	Increase in Runx2, OCN and ALP for osteoblast differentiation	Activated MAPK (ERK, JNK, p38) signaling pathway, suppress NFATC1/c-Fos signaling	Pre-clinical	Promote osteogensis, inhibit osteoclastogenesis	[16]
	*Epimedium brevicornu*	Icariin (ICA)	Thioacetamide induced bone loss	Suppress RANKL-p38/ERK-NFAT signaling	Pre-clinical	Inhibit osteoclast differentiation	[19]
	*Rabdosia rubens*	Oridonin (ORI)	Increase ALP, Ca^2+^ nodules, inhibit TRAP, NFATc1/c-Fos	Activated Wnt/β-catenin signaling	Pre-clinical	Promote osteogenesis, inhibit osteoclastogenesis	[20]
	*Viburnum lutescens*	Hydroxyurosolic acid (HUA)	Inhibits osteoclast differentiation	Suppress c-Fos and NF-κB signaling	Pre-clinical	Inhibit osteoclast differentiation	[21]
	*Tripterygium wilfordii*	Triptolide	Increased OPG, decreased RANKL	Activated OPG/RANKL signaling	Pre-clinical	Inhibit osteoclastogenesis	[22]
	*Acorus tatarinowii*	α-asarone (ASA)	Inhibit osteoclastogenesis	AKT, p38 and NF-κB, followed by NFATc1/c-fos signaling pathway	Pre-clinical	Attenuate osteoclastogenesis	[23]
TRAP	*Drynariae fortunei*	Naringenin (NAR)	Increased ALP, OCN, Runx2, parathyroid receptor1 (PTH1R), MSC proliferation	Activated Wnt/β-catenin signaling	Pre-clinical	Influence osteocyte/osteoblast/osteoclast activity	[24]
	Fruits	Resveratrol (RSV)	Increased ALP, Runx2, SOD, PINP SIRT1, Osx decreased TRAP	Activated SIRT1/FOXO1 signaling	Pre-clinical	Promote osteogenesis	[25]
MMP9	*Enicosttema axillare*	Swertiamarin	Inhibit cytokines, Cox2, MMPs and RANKL	Activated MAPK (ERK, JNK, p38) signaling	Pre-clinical	Amoliate inflammation, attenuate osteoclastogenesis	[26]
NLRP3	*Drynaria fortunei*	Dioscin	Increased ALP, Runx2, OCN	Suppress NLRP3	Pre-clinical	Promote osteogenesis	[27]
		Extract	Suppress NLRP3 cytokines and Notch1 by increasing SIRT1	Suppress Notch1 independent of SIRT1	Pre-clinical	Amoliate inflammation, improve lipid profile in OP	[28]
Immune cells and Cytokines	*Epimedium brevicornu*	Baohuoside 1	Increased ALP, MSC differentiation	Inhibit cytokines and adipogenesis	Pre-clinical	Promote BMSC differentiation	[29]
PPAR-γ	*Epimedium brevicornu*	Icariin (ICA)	Increased ALP, TGF-β1 and OPG, decreased RANK expression and NF-kβ	PPAR-γ inhibition	Pre-clinical	Promote and/or inhibit differentiation of stem cells based on osteoblast-osteoclast co-cultering	[30]
VEGF	*Rhodiola rosea*	Salidroside (SAL)	Increased VEGF	Activated HIF-1α/VEGF signaling	Pre-clinical	Promotes osteogenesis and angiogenesis	[31]
BMP2	*Mangifera indica*	Mangiferin	Increased OPG	Activated BMP (Smad1/5/8, Smad4) signaling	Pre-clinical	Inhibit MC3T3 cells apoptosis	[32]
	*Rhodiola rosea*	Salidroside	Increased phosphorylation of BMP2, BMP7, Smad 1/5/8 and ERK1/2	Activated BMP signaling	Pre-clinical	Stimulate osteoblast differentiation	[33]
	*Epimedium brevicornu*	Extract	Enhanced expression of Runx2, Osx and BMP2	Activated AC10/cAMP/PKA/CREB	Pre-clinical	Increases maximum bone density during growth	[34]
Osx	*Helminthostachys zeylanica*	Ugonin K	Increased Osx and runx2	Activated MAPK (ERK, JNK, p38) signaling	Pre-clinical	Stimulate osteoblast differentiation	[17]
Cathepsin K	*Polygonatum sibiricum*	Polysaccharide (PSP)	Increased ALP, runx2, Col1A1, OCN, supresss ACP5 and cathepsin K	Suppress cathepsin K	Pre-clinical	Promotes osteogenesis, inhibit osteoclastogenesis	[35]
	*Drynariae fortunei*	Extract	Increased ALP, BMP2	Suppress resorption induced by Catk	Pre-clinical	Promote osteoblast differentiation	[36]
Oxidative stress (OS)	*Curculigo orchioides Gaertn*	Orcinol glucoside (OG)	Decrease oxidative stress and autophagy of osteoclast	Activated Nrf2/Keap1 and mTOR signaling	Pre-clinical	Promote bone health by reducing oxidative stress	[37]
	*Moringa oleifer*	Extract	Osteoblast survival	PI3K, AKT, Fox1 signaling	Pre-clinical	Promote osteoblast differentiation	[38]
	*Cullen corylifolium*	Corylifol A	Increased antioxidant enzymes; CAT, HMOX1, NQO1	Suppress ROS and osteoclast differentiation	Pre-clinical	Inhibit osteoclastogenesis	[39]
	*Eurya cilliata*	Quercetin (QUE)	Increased ALP, Ca^2+^ and collagen	Activated MAPK (ERK, JNK, p38) signaling	Pre-clinical	Enhance osteoblast activity	[40]
ROS	*Crocus sativus*	Crocin (CRO)	Increased Bcl2, bax and cytochrome C	Suppress ROS/Ca2 mitochrondrial signaling	Pre-clinical	Protect osteoblast from ROS	[41]
	*Radix ophiopogon japonicus*	Ophiopogonin D (OP-D)	Decreased ROS, CTX1 and TRAP	FOXO3a-β-catenin signaling	Pre-clinical	Promotes osteogenesis/inhibit osteoclastogenesis	[42]
	Grapes	Resveratrol (RSV)	Decreased ROS and RANKL	Activated Wnt/β-catenin signaling	Pre-clinical	Inhibit RANKL-induced osteoclastogenesis	[43]
	*Polygonum multiflorum*	Tetrahydroxystilbene-2-O-β-D-glucoside (TSG)	Increased ALP, OCN while decreased RANKL, ROS, malondialdehyde (MDA)	Activated JNK, PI3K/AKT and ROS-NO signaling	Pre-clinical	Protect osteoblast	[44]
	*Epimedium brevicornu*	Icariin (ICA)	Increased BMP2 and cAMP	cAMP/PKA/CREB signaling	Pre-clinical	Promote osteogenesis	[45,46]
adipocyte differentiation	*Coptischinensis franch*	Berberine (BBR)	Increased OCN, Runx2, Cox2	Activated MAPK (ERK, JNK, p38) signaling, suppress adipocyte differentiation	Pre-clinical	Promote osteoblast differentiation	[47]
Inflammation	*Curcuma longa*	Curcumin (CUR)	Decreased cytokines, promote bone healing	Bone composite mixing PMMA	Pre-clinical	Promote bone healing	[48]
	*Eucalyptus globules*	Eucalyptol (EU)	Increased ERK phosphorylation	Activated MAPK (ERK, JNK, p38) signaling	Pre-clinical	Promote osteoblast differentiation	[49]
NF-κB	*Curcuma longa*	Curcumin (CUR)	Inhibit TNF-α and IL-17	suppress NF-κB signaling	Pre-clinical	Anti-inflammatory and anti-apoptosis effect	[50]
	*Salvia miltiorrhiza*	Tanshinone IIA	Inhibit expression of TRAF6 and NFTAc1	suppress NF-κB signaling	Pre-clinical	Reduce inflammation and pain in bone tumor	[14]
	Citrus fruits	Naringin (NAG)	Increased BMP2	Activated PI3K, AKT, c-Fos/c-Jun and AP1 signaling	Pre-clinical	Promote osteogenesis/inhibit osteoclastogenesis	[51,52]
Macrophage M1 polarization	*Psoralea corylifolia*	Isobavachalcone (ISO)	Decreased inflammatory cytokines and macrophage M1 polarization	Suppress ERK, NF-κB signaling	Pre-clinical	Reduce osteoclast activity and bone resorption	[53]
	*Polygonum aviculare*	Avicularin (AL)	Decreased inflammatory cytokines and macrophage M1 polarization	Suppress NF-κB signaling	Pre-clinical	Reduce inflammation and M1 macrophage activity	[54]
Smad degradation	*Drynariae fortunei*	Indirubin	Inhibit Smad	BMP signaling	Pre-clinical	Regulate bone formation	[55]
Epigenetics and autophagy	Sprouts of cruciferous vegetables	Sulforaphane	Prevent osteoblast apoptosis and promote its differentiation	Activated gene expression by TET1/2 dependent DNA	Pre-clinical	Promote osteogenesis, inhibit osteoclastogenesis via epigenetic mechanism	[56]
	*Carthamus tinctorius*	Hydroxysafflor yellow A (HSYA)	Regulate β-catenin and promote MSC differentiation to osteoblast	Histone demethylation by KDM7A	Pre-clinical	Promote osteogenesis by epigenetically regulating β-catenin	[57]
	*Rheum palmatum*	Rhein and its derivatives	Estrogen dependent bone loss and formation	Decreased Dnmt1/dnm3a activity and hydroxymethylation of Klotho promoter	Pre-clinical	Reverse DNA methylation that activate Klotho gene and promote bone health	[58]
	*Ribes nigrum*	Anthocyanins	Enhanced proliferation of osteoblasts	Altered expression of Bcl2, Bax, HDAC1/3 by upregulated SIRT1/3 and PGC-1α	Pre-clinical	Promote osteogenesis	[59]
	Grapes	Resveratrol (RSV)	Induce SIRT1 deacetylase	Inhibit NF-κB Signaling	Pre-clinical	Inhibit osteoclastogenesis by influencing P300 interating with RANKL	[60]
	*Reseda odorata*	Luteolin	Bcl2, Bax, caspase 3/9	Regulate ERK/LRP-5/GSK-3β signaling	Pre-clinical	Enhance osteogenesis, inhibit osteoclastogenesis	[61]

**Table 2 pharmaceuticals-17-00984-t002:** Patents of herbal compositions used for treating osteoporosis.

EAHM	Patent	URLs
Extract of *Epimedii* and *Salvia miltiorrhiza*	CN101023982A	https://patents.google.com/patent/CN102218089B/en, accessed on 22 July 2024
Extract of *Drynariae*	WO2002053164	http://engpat.kipris.or.kr/engpat/biblioa.do?method=biblioFrame, accessed on 22 July 2024
Extract of *Epimedii* alongwith other active components	CN117949573A	https://patents.google.com/patent/CN117949573A/en?oq=CN117949573A, accessed on 22 July 2024
Extracts of *Epimedii*, *Drynariae* and *Achyranthis Bidentatae*	CN103251671B	https://patents.google.com/patent/CN103251671B/en?oq=CN103251671B, accessed on 22 July 2024
Extracts of *Epimedii*, *Polygonum multiflorum*	CN105616630A	https://patents.google.com/patent/CN105616630A/en?oq=CN105616630A, accessed on 22 July 2024
Extracts of *Epimedii*, *Polygonum multiflorum* and *Drynariae*	CN105456711A	https://patents.google.com/patent/CN105456711A/en?oq=CN105456711A, accessed on 22 July 2024

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
