# Peer review of "Emerging Roles of Natural Compounds in Osteoporosis: Regulation, Molecular Mechanisms and Bone Regeneration"

_pharmaceuticals, 2024, doi:10.3390/ph17080984_

Round 1

Reviewer 1 Report

Comments and Suggestions for Authors
  • Improvement of Figure 1: To enhance the clarity of Figure 1, I recommend the following changes:

    • a. Use a green circle to represent the compound.
    • b. Increase the links between each pathway and adjust the arrows to make the connections more logical and straightforward.
    • c. The current figure is too simplistic and does not fully convey the message. A clearer figure should follow the order of Section 2 in the manuscript, including ARNKL, BMP, MAPK, etc., to avoid potential confusion for the readers.
  • Reorganization of Figures: Based on the paper's logic, I suggest introducing a new Figure 1 that summarizes the roles of TCM (Traditional Chinese Medicine) in bone regeneration. The current Figure 1 should then be moved to Figure 2. Please include this additional figure to better illustrate the overarching concepts.

  • Expansion of Section 1.6: It is highly recommended to expand Section 1.6 to discuss the balance between osteoclast (OC) survival and osteoblast (OB) function in bone regeneration. The current text may lead readers to undervalue the importance of maintaining OC survival in bone regeneration. A more detailed explanation of the relationship between OC and OB would enhance the readers' understanding of this balance.

  • Citation of Tables and Figures: Please ensure that all tables and figures are cited correctly within the manuscript. Specifically, Table 2 needs to be referenced, and Table 1 appears to be missing from the manuscript. Proper citation and inclusion of all tables and figures are essential for clarity.

  • Details in Table 2: In Table 2, it would be beneficial to include significant findings, such as whether RSV inhibits or activates NF-κB. Including these details will provide readers with quick and useful information and is a valuable addition for a review.

  • Inclusion of Representative Works: It is suggested to include a montage of representative work related to TCM. This visual representation can help illustrate key points and provide a comprehensive overview of relevant studies.

  • Section Descriptions: It is recommended to include a brief description at the beginning of each section to explain the purpose of the section and its relevance. This will help guide readers through the content and improve the overall structure of the manuscript.

Comments on the Quality of English Language

language is understandable 

Author Response

We have updated the term Traditional Chinese medicine (TCM) as it is limited to only Chinese medicines, therefore we have used much broader term “East Asian Herbal Medicine (EAHM)” in our manuscript which covers all herbal medicines in East Asian countries such as China, Taiwan, Korea and Japan.

Reviewer 1

Improvement of Figure 1: To enhance the clarity of Figure 1, I recommend the following changes:

  1. Use a green circle to represent the compound.

We have used green circles to represent the compounds

  1. Increase the links between each pathway and adjust the arrows to make the connections more logical and straightforward.

We have now made more connections and adjusted the arrows to make figure more logical and straightforward.

  1. The current figure is too simplistic and does not fully convey the message. A clearer figure should follow the order of Section 2 in the manuscript, including ARNKL, BMP, MAPK, etc., to avoid potential confusion for the readers.

We have prepared a new Figure 1

Reorganization of Figures: Based on the paper's logic, I suggest introducing a new Figure 1 that summarizes the roles of TCM (Traditional Chinese Medicine) in bone regeneration. The current Figure 1 should then be moved to Figure 2. Please include this additional figure to better illustrate the overarching concepts.

We have now reorganized the figures as suggested by the reviewer.

Expansion of Section 1.6: It is highly recommended to expand Section 1.6 to discuss the balance between osteoclast (OC) survival and osteoblast (OB) function in bone regeneration. The current text may lead readers to undervalue the importance of maintaining OC survival in bone regeneration. A more detailed explanation of the relationship between OC and OB would enhance the readers' understanding of this balance.

The section 1.6 has been extended.

Loss of Bcl2 leads to upregulated caspase3 expression, increased apoptosis in osteoclasts via the mitochondrial apoptotic pathway. The most prevalent communicator between osteoclasts and osteoblasts are the osteocytes which promote a well-organized and effective bone remodeling. Upon osteocyte apoptosis, ATP is released through the Panx1 channels which can increase RANKL expression as well as promote fusion of osteoclast precursor cells into mature osteoclast. RANKL and TNF-α in osteoclast enhances BclxL expression to prevent bisphosphonate-induced apoptosis. It has been reported that luteolin from Reseda odorata can attenuate glucocorticoid-induced osteoporosis by regulating ERK/LRP-5/GSK-3β signaling pathway in vivo and in vitro [46]. Similarly, mangiferin inhibits apoptosis in dexamethasone-induced MC3T3-E1 cell lines by activating BMP2/SMAD1 signaling and altering RANKL and OPG levels [47]. Additional studies revealed that overexpression of Bcl2 disrupted osteoblast development, impairing the cell's capacity to develop and generate bone. It also resulted in the loss of osteocytes, which are significant adult bone cells [48]. The role of Bcl2 subfamily proteins in osteoporosis is very complex and still in infancy since more research is needed. Quercitrin nanocoated implant dramatically reduced the expression of genes linked to osteoclasts, such as Trap, RankL, Ctsk, ATPase and Mmp9, in vivo [49].

Citation of Tables and Figures: Please ensure that all tables and figures are cited correctly within the manuscript. Specifically, Table 2 needs to be referenced, and Table 1 appears to be missing from the manuscript. Proper citation and inclusion of all tables and figures are essential for clarity.

We apologize for this. We have now correctly cited all figures and tables within the manuscript.

Details in Table 2: In Table 2, it would be beneficial to include significant findings, such as whether RSV inhibits or activates NF-κB. Including these details will provide readers with quick and useful information and is a valuable addition for a review.

Thanks for the valuable suggestion. We have now included in Table 1 (mentioned incorrectly as table 2) that RSV inhibits the activation of NF-κB signaling.

Inclusion of Representative Works: It is suggested to include a montage of representative work related to TCM. This visual representation can help illustrate key points and provide a comprehensive overview of relevant studies.

Thanks for the suggestion. A montage of representative work related to TCM is added.

Section Descriptions: It is recommended to include a brief description at the beginning of each section to explain the purpose of the section and its relevance. This will help guide readers through the content and improve the overall structure of the manuscript.

Thanks for the suggestion. We have already included brief description at the beginning in the Introduction section.

Reviewer 2 Report

Comments and Suggestions for Authors

The Article “Emerging Roles of Natural Compounds in Osteoporosis: Regulation, Molecular Mechanisms and Bone Regeneration” the potential of Traditional Chinese Medicine (TCM) to enhance bone health, particularly in combating osteoporosis, a significant global challenge. It explores the complex mechanisms by which TCM compounds influence bone regeneration, focusing on the balance between osteogenesis (bone formation) and osteoclastogenesis (bone resorption), and the crucial signalling pathways involved in bone remodelling. The review underscores TCM's anti-inflammatory effects within the bone microenvironment, promoting osteoblast viability and regulating bone turnover markers. It also delves into epigenetic modifications, revealing how TCM impacts DNA methylation and histone changes to orchestrate bone regeneration. The comments are as follows:

1.      The limitation of the review is the limited representation of work in illustration and tabular form.

2.      I don’t find table 1, directly there is table 2.

3.      Table 2. Role of Natural Chinese medicine and their anti-osteoporotic effects. Is this data from preclinical or clinical? Add a column on the study performed and the outcome of it.

4.      Add patents and clinical trials or case reports to support the review in tabular summarization.

5.      Section 6, is very general, please add a citation and write it critically with findings.

6.      Line 169, 261, 411, 465: (Luo et al., 2018; Tabatabaei-Malazy et al., 169 2017), (D. W. 261 Lee et al., 2023), (Shakibaei et al., 2011), (Deng et al., 2020), wrong ref style. Please check the full manuscript for such errors.

7.      Marketed formulations need to be included.

8.      Report the toxicity or side effects in the challenge part.

Author Response

We have updated the term Traditional Chinese medicine (TCM) as it is limited to only Chinese medicines, therefore we have used much broader term “East Asian Herbal Medicine (EAHM)” in our manuscript which covers all herbal medicines in East Asian countries such as China, Taiwan, Korea and Japan.

Reviewer 2

  1. The limitation of the review is the limited representation of work in illustration and tabular form.

Thanks for the suggestions. We have added more illustrations and tables.

  1. I don’t find table 1, directly there is table 2.

We apologize for this mistake. Table number has been changed to Table 1.

  1. Table 2. Role of Natural Chinese medicine and their anti-osteoporotic effects. Is this data from preclinical or clinical? Add a column on the study performed and the outcome of it.

We have added a column study performed and outcome in Table 1

  1. Add patents and clinical trials or case reports to support the review in tabular summarization.

We have added patents to support the review in tabular summarization as Table 2.

  1. Section 6, is very general, please add a citation and write it critically with findings.

We have added more details and citations in section 6.

  1. Line 169, 261, 411, 465: (Luo et al., 2018; Tabatabaei-Malazy et al., 169 2017), (D. W. 261 Lee et al., 2023), (Shakibaei et al., 2011), (Deng et al., 2020), wrong ref style. Please check the full manuscript for such errors.

We have checked the whole manuscript for such errors and the mentioned Lines 169, 261, 411, 465 have been corrected now according to Journal’s format.

  1. Marketed formulations need to be included.

We have included the marketed formulations in the section 3.6

  1. Report the toxicity or side effects in the challenge part.

We have included in the section 6.4

Round 2

Reviewer 1 Report

Comments and Suggestions for Authors

thank you for your revise

Comments on the Quality of English Language

good

Reviewer 2 Report

Comments and Suggestions for Authors

Thank you for revising as per suggestions. I am satisfied with this version.